# Episodic $N_2O$ emissions following tillage of a legume-grass cover crop mixture

Alison Bressler[1], Jennifer Blesh[1]

[1]School for Environment and Sustainability, University of Michigan, Ann Arbor, 48104, USA

*Correspondence to*: Alison Bressler (asbressl@umich.edu)

**Abstract.** Nitrogen (N) fertilizer inputs to agricultural soils are a leading cause of nitrous oxide ($N_2O$) emissions. Legume cover crops are an alternative N source that can reduce agricultural $N_2O$ emissions compared to fertilizer N. However, our understanding of episodic $N_2O$ flux following cover crop incorporation by tillage is limited and has focused on single species cover crops. Our study explores whether increasing cover crop functional diversity with a legume-grass mixture can reduce

pulse emissions of $N_2O$ following tillage. In a field experiment, we planted crimson clover (*Trifolium incarnatum L.*), cereal rye (*Secale cereal L.*), a clover-rye mixture, and a no-cover control at two field sites with contrasting soil fertility properties in Michigan. We hypothesized that $N_2O$ flux following tillage of the cover crops would be lower in the mixture and rye compared to the clover treatment, because rye litter can decrease N mineralization rates. We measured $N_2O$ for approximately two weeks following tillage to capture the first peak of $N_2O$ emissions in each site. Across cover crop treatments, the higher

fertility site, *CF*, had greater cover crop biomass, twofold higher aboveground biomass N, and higher cumulative $N_2O$ emissions than the lower fertility site, *KBS* ($413.4 \pm 67.5$ g $N_2O$-N ha$^{-1}$ vs. $230.8 \pm 42.5$ g $N_2O$-N ha$^{-1}$; $P = 0.004$). There was a significant treatment effect on daily emissions at both sites. At *CF*, $N_2O$ fluxes were higher following clover than the control 6 days after tillage. At *KBS*, fluxes from the mixture were higher than rye 8 and 11 days after tillage. When controlling for soil fertility differences between sites, clover and mixture led to approximately twofold higher $N_2O$ emissions compared to rye and

fallow treatments. We found partial support for our hypothesis that $N_2O$ would be lower following incorporation of the mixture than clover. However, treatment patterns differed by site, suggesting that interactions between cover crop functional types and background soil fertility influence $N_2O$ emissions during cover crop decomposition.

## 1 Introduction

Nitrogen (N) losses from grain agroecosystems contribute to climate change through nitrous oxide ($N_2O$) emissions (Robertson

and Vitousek, 2009). Globally, $N_2O$ emissions from agricultural soils increased by 11% from 1990 to 2005 and are projected to increase by another 35% between 2005 and 2030 (USEPA, 2012). In the U.S., approximately 75% of $N_2O$ emissions come from agricultural soils (USEPA, 2021), and the amount of N added to soil from synthetic fertilizers is the primary driver of these high emissions (Millar et al., 2010; Han et al., 2017; Eagle et al., 2020). Generally, total N inputs are correlated with N losses from agroecosystems (Robertson and Vitousek 2009). However, diversified grain rotations with legume N sources,

which add biologically fixed $N_2$ to fields, better balance N inputs with harvested exports and have lower potential for N losses compared to synthetic fertilizers (Drinkwater et al., 1998; Blesh and Drinkwater, 2013; Robertson et al., 2014). Legumes can be added to rotations as overwintering cover crops, which are unharvested crops planted in the fall and terminated in the spring in temperate regions. As an organic N source, legume litter supplies organic substrates to support microbial processes that can increase soil organic matter (SOM) pools and N retention in SOM (Drinkwater et al., 1998; Syswerda et al., 2012; Blesh and

Drinkwater, 2013). Further, diversified rotations with legume N sources could reduce or replace the use of synthetic N fertilizers, thereby reducing greenhouse gas emissions associated with fertilizer production and application (Norskov and Chen, 2016).

Two key factors that affect $N_2O$ emissions are soil disturbance through tillage and crop functional traits (Gelfand et al. 2016).

In agroecosystems, even small increases in crop functional diversity (e.g., 2-3 species cover crop mixtures with complementary traits) can substantially impact ecosystem functions such as SOM accrual, N cycling processes, and weed suppression (Drinkwater et al., 1998; McDaniel et al., 2014; Tiemann et al., 2015; Blesh, 2017). For example, the timing and rate of N release from different cover crop functional types (i.e., C4 vs C3 grasses, N fixing legumes) during decomposition affects the potential for N losses (Millar et al., 2004; White et al., 2017) through effects on soil N availability. Interactions between the

biochemical composition of fresh litter inputs and background soil properties, including the microbial community, are key drivers of microbial decomposition dynamics and N mineralization rates (Cheng, 2009; Kallenbach et al., 2019). Consequently, legume cover crops, which have a high N concentration, may result in higher production of $N_2O$ after disturbances like tillage compared to cover crops that include non-legume species (Alluvione et al., 2010; Huang et al., 2004; Millar et al., 2004; Gomes et al., 2009). The effects of litter C:N on N mineralization and $N_2O$ flux may be particularly evident when comparing sole

legumes with lower ratios (e.g., C:N < 15) to grass cover crops with higher C:N (e.g., > 30) (Baggs et al., 2003). For example, prior research on legume-grass mixtures revealed that they reduced N leaching compared to sole legumes, while enhancing N supply compared to sole grasses, providing multiple ecosystem functions (Kaye et al., 2019). However, there is limited data on $N_2O$ losses following cover crops in organically managed agroecosystems, and the effects of mixtures of complementary functional types on $N_2O$ emissions are poorly understood.


Understanding the timing of $N_2O$ emissions is also key to reducing N losses from crop rotations (Wagner-Riddle et al., 2020). Millar et al. (2004) found that $N_2O$ fluxes are episodic in a cropping system with corn and legume cover crops as the sole source of new N. Specifically, 65-90% of $N_2O$ emissions occurred during the first 28 days following tillage of legume cover crops, over an 84-day measurement period. Similarly, Gomes et al. (2009) found greater $N_2O$ emissions during the first 45

days after terminating cover crops with a roller cutter and herbicide compared to the rest of the year. Gelfand et al. (2016) observed high temporal variability in $N_2O$ fluxes measured for 20 years in different temperate grain cropping systems and suggested that emissions following tillage were a primary driver of this variation in the two agroecosystems with cover crops. Therefore, there is a need to measure $N_2O$ in the weeks following cover crop termination to understand pulse $N_2O$ fluxes,

particularly when legumes are the sole, or primary, source of N additions. Further, to our knowledge no studies have tested
whether legume-grass mixtures reduce pulse $N_2O$ during this critical period compared to sole-legume cover crops.

Variability in soil conditions also plays an important role in soil $N_2O$ flux. Edaphic characteristics, such as soil texture (Gaillard et al., 2016), soil organic carbon (SOC) (Bouwman et al., 2002; Dhadli et al., 2016), and interannual rainfall patterns can often explain more variation in $N_2O$ emissions than treatment differences (Basche et al., 2014; Ruser et al., 2017). One study with
synthetic N fertilizer additions on clayey Oxisols in Brazil found higher $N_2O$ losses from more intensively managed fields with lower labile SOM fractions and total C content (de Figueiredo et al., 2017). In fields with organic N sources, SOM fractions with relatively short turnover times (i.e., years to decades) likely influence N mineralization following cover crop incorporation and resulting $N_2O$ emissions. Free particulate organic matter (fPOM) and occluded particulate organic matter (oPOM), which is physically protected in soil aggregates, are both indicators of nutrient cycling capacity in soil (Marriott and Wander, 2006).
Prior studies have found that POM N concentrations are positively correlated with potential N mineralization rates (Blesh, 2019), and that this relationship varies with soil texture and management history (Luce, 2016). It is therefore critical to assess $N_2O$ emissions in soils with different properties, such as SOM, POM, and nutrient stocks, which reflect the environmental context and land management histories.

In this field experiment, we tested the effects of a legume-grass cover crop mixture on agroecosystem N cycling processes compared to either species grown alone during the first flux of $N_2O$ following tillage. The experiment was conducted at two sites in Michigan with contrasting soil fertility properties. Our specific objectives were to: (1) quantify cover crop functional traits, including C:N and legume N inputs from biological N fixation (BNF) and (2) test the effects of cover crop treatment on pulse $N_2O$ fluxes following spring tillage, when emissions are expected to be greatest in agroecosystems that rely on legume
N sources. Our hypothesis was that the legume-grass mixture would result in lower pulse $N_2O$ fluxes than the sole-planted legume due to a higher C:N and a smaller new N input to soil from BNF.

## 2 Materials and Methods

### 2.1 Site description and experimental design

The study was conducted on two sites in two regions of Michigan, USA. The first site (*CF*) was located at the University of
Michigan's Campus Farm (Lat/Long: N 42° 17' 47", W 83° 39' 19" Elevation: 259.08 m), was previously in a grass fallow with periodic mowing for over 45 years. The experiment at *CF* was conducted in the 2017-2018 overwintering cover crop season. The site resides on a glacial till plain with well drained sandy loam soils in the Fox series which are mixed, superactive, mesic Typic Hapludalfs. The soil had 2.5% organic matter, 21.5% clay, and a pH of 6.35. The site received 1030 mm of rainfall during the experiment (August 2017 – September 2018) with an average temperature of 10.2 °C. The second site (*KBS*) was

located in the biologically-based cropping system in the Main Cropping System Experiment (MCSE) of the Kellogg Biological Station Long-Term Ecological Research site (Lat/Long: N 42° 14' 24", W 85° 14' 24" Elevation: 288 m). The field has been in a corn-soy-winter wheat rotation managed using organic practices for over 30 years. The experiment at *KBS* was conducted in the 2019-2020 overwintering cover crop season. This site is on a glacial outwash plain with well drained loam, sandy loam, and sandy clay loam soils in the Kalamazoo and Oshtemo series which are mixed, mesic Typic Hapludalfs (Crum and Collins, 1995). The soil had 1.74% organic matter, 19.4% clay, and a pH of 6.59. The site receives an average of 933 mm $yr^{-1}$ with an average temperature of 9.2 °C. Neither field received any fertilizer or manure applications before or during the experiment.

In a randomized complete block design, we planted four cover crop treatments in 4.5 x 6 m plots at *CF:* (1) cereal rye (seeding rate: 168 kg $ha^{-1}$), (2) crimson clover (seeding rate: 34 kg $ha^{-1}$), (3) clover-rye mixture (seeding rate: 67 kg $ha^{-1}$ rye, 17 kg $ha^{-1}$ clover) (4) and a weedy fallow control, in four blocks by broadcasting seed on 16 August 2017. We planted four cover crop treatments into 3.1 x 12.2 m plots at *KBS*: (1) cereal rye (seeding rate: 100.9 kg $ha^{-1}$), (2) crimson clover (seeding rate: 16.8 kg $ha^{-1}$), (3) clover-rye mixture (seeding rate: 50.4 kg $ha^{-1}$ rye, 9.0 kg $ha^{-1}$ clover) (4) and a weedy fallow control, in four blocks with a grain drill on 31 July 2019. Seeding rates were reduced for the site planted with a grain drill due to higher likelihood of germination. The cover crops overwintered and were rototilled into the soil on 24 May 2018 (*CF*) and on 26 May 2020 (*KBS*) followed by corn planting on 14 June 2018 (*CF*) and on 1 June 2020 (*KBS*). Cover crops had 4,501 growing degree days at *KBS* and 3,898 at *CF*.

### 2.2 Baseline Soil Sampling

Prior to planting, we collected a composite, baseline soil sample from each replicate block at *CF*, and from each treatment plot within each replicate block at *KBS*, to determine initial soil conditions and characterize soil fertility status at both experimental sites. In each plot, we estimated bulk density from the fresh mass of 10 composited soil cores (2 x 20 cm) and adjusted for soil moisture, determined gravimetrically. Subsamples of ~ 50 g were also analysed for soil texture using the hydrometer method. Air-dried soil was mixed and soaked with 100 mL of sodium hexametaphosphate and blended for 5 min. The mixture was transferred to a glass sedimentation cylinder and filled to 1L with tap water. The slurry was mixed with a metal plunger and hydrometer readings were taken 40 seconds and 2 hours after the plunger was removed. Percent sand was calculated from the 40 second reading and percent clay from the 2-hour reading.

At sampling, we sieved a subsample of fresh soil to 2 mm and measured extractable and potentially mineralizable N in triplicate for each soil sample. We immediately extracted inorganic N ($NO_3^-$ + $NH_4^+$) in 2 mol/L KCl. The amount of $NO_3^-$ + $NH_4^+$ in each sample was analysed colorimetrically on a discrete analyser (AQ2; Seal Analytical, Mequon, WI). We also performed a 7-day anaerobic N incubation and then extracted $NH_4^+$ in 2 mol/L KCl. Soil weights for extractions and incubations were adjusted for soil moisture. Potentially mineralizable N (PMN) was calculated by subtracting the initial amount of $NH_4^+$ in the soil from the $NH_4^+$ released during the 7-day incubation (Drinkwater et al., 1996).

Particulate organic matter (POM) (> 53 μm) was separated from triplicate 40-g subsamples of unsieved, air-dried soil based
on size and density (Marriott and Wander, 2006; Blesh, 2019). To isolate the light fraction POM (also called free POM or
fPOM), the subsamples were gently shaken for 1 hour in sodium polytungstate (1.7 g/cm3), allowed to settle for 16 hours, and
free POM floating on top of the solution was removed by aspiration. To separate the physically protected, or occluded, POM
fraction (oPOM), the remaining soil sample was shaken with 10% sodium hexametaphosphate to disperse soil aggregates and
then rinsed through a 53-μm filter (Marriott and Wander, 2006). Protected POM was then separated from sand by decanting.
The C and N of both POM fractions (fPOM and oPOM) were measured on an ECS 4010 CHNSO Analyzer (Costech Analytical
Technologies, Valencia, California, USA). Total soil C and N (to 20 cm) were measured by dry combustion on a Leco TruMac
CN Analyzer (Leco Corporation, St. Joseph, Michigan, USA) (Blesh, 2019).

## 2.3 Aboveground biomass sampling and analysis

We sampled aboveground biomass from all treatments on 22 May 2018 (*CF*) and on 26 May 2020 (*KBS*), from one 0.25 m$^2$
quadrat randomly placed in each plot, avoiding edges. Shoot biomass was cut at the soil surface, separated by species (with
weeds grouped together), dried at 60 °C for 48 hours, weighed, and coarsely ground (< 2 mm) in a Wiley mill. We analysed
the biomass for total C and N by dry combustion on a Leco TruMac CN Analyzer (Leco Corporation, St. Joseph, MI).

## 2.4 Legume N fixation by natural abundance

We estimated BNF by crimson clover using the natural abundance method (Shearer & Kohl, 1986). Shoot biomass from the
clover in monoculture and mixture and rye in monoculture (the non-N$_2$ fixing reference plant), were collected in the field,
dried, weighed, and finely ground (<0.5 mm). Samples were analyzed for total N and δ$^{15}$N enrichment using a continuous flow
Isotope Ratio Mass Spectrometer at the UC Davis Stable Isotope Facility. The percent N derived from the atmosphere (i.e.,
%Ndfa) was calculated using the following mixing model Eq. (1):

$$\text{\%Ndfa} = 100 \times ((\delta^{15}N_{ref} - \delta^{15}N_{legume}) / (\delta^{15}N_{ref} - B)) , \tag{1}$$

where δ$^{15}$N$_{ref}$ is the δ$^{15}$N signature of the reference plant (rye), δ$^{15}$N$_{legume}$ is the δ$^{15}$N signature of the clover and B is defined as
the δ$^{15}$N signature of a legume when dependent solely on atmospheric N$_2$. B values were determined by growing crimson
clover species in the greenhouse in a N-free medium following methods in Blesh (2017). After conducting two B-value
experiments with crimson clover (one per site), we found a mean B-value of -1.57, which we used in our calculation of %Ndfa.
We estimated BNF (kg N ha$^{-1}$) by multiplying field values for aboveground biomass by shoot % N, and then by %Ndfa. The
natural abundance method is generally considered reliable when the δ$^{15}$N signature of the legume and reference plants are
separated by 2 ‰ (Unkovich et al., 2008). At the *KBS* site, this criterion was met; however, we did not find adequate separation
between the legume and reference species at *CF*. We therefore estimated BNF at *CF* using the mean %Ndfa values from *KBS*

for clover in mixture and monoculture. Given this, we also conducted a sensitivity analysis to determine how variation in %Ndfa at *CF* would affect model outcomes.

### 2.5 N$_2$O flux following soil disturbance

We used the static chamber method (Kahmark et al., 2018) to measure the first pulse of N$_2$O emissions in each field following tillage of all experimental plots. All measurements occurred between 9 am and noon. At *CF*, we measured N$_2$O once before and five times after cover crop incorporation over 18 days. At *KBS*, we measured N$_2$O seven times after cover crop incorporation over 15 days. These periods captured the main episode of N$_2$O flux following tillage and initial decomposition of cover crop residues. During the N$_2$O measurement period, each site received the same amount of precipitation (15 mm) and had the same average temperature (20.6 °C).

Static chambers at *KBS* were made from stainless steel cylinders (diameter: 28.5 cm) and chambers at *CF* were made from Letica 3.5-gallon pails with the bottom removed to create a cylinder (diameter at top: 28.5 cm, diameter at bottom: 26 cm). Chamber lids were fitted with O-ring seals to create an airtight container during sampling. Each lid was equipped with a rubber septa port for extraction of gas samples. Before each sampling date, static chambers were installed in the ground and allowed to rest for at least 24 hours to reduce the impact of soil disturbance on measured emissions. The morning before each sampling event, the depth from the lip of the chamber to the ground was measured at three locations inside the chamber to calculate the internal volume. Lids were then placed securely on the chamber and 10 mL samples were extracted using a syringe every 20 minutes over a period of 60 minutes. Each 10 mL sample was stored, overpressurized, in a 5.9 mL, graduated glass vial with an airtight rubber septum (Labco Limited, Lampeter, UK). We analysed samples for N$_2$O using a gas chromatograph equipped with an electron capture detector (Agilent, Santa Clara, CA). N$_2$O flux was calculated as the change in headspace N$_2$O concentration over the 60-minute time-period. Each set of 4 data points (0, 20, 40, and 60 minutes) were analysed using linear regression and screened for non-linearity.

### 2.6 Soil inorganic nitrogen sampling

On the day after tillage, and again 12-13 days later, we measured soil inorganic N (NH$_4^+$ + NO$_3^-$) near the static chambers at both sites. We collected four to six, 2 cm diameter soil cores to 10 cm depth, within 1 m of each static chamber. Samples were immediately homogenized, sieved to 2 mm, extracted with 2 M KCl, and analysed for soil moisture using the gravimetric method. Extractions were stored at -20 °C and later thawed and analysed for NO$_3^-$ and NH$_4^+$ colorimetrically on a discrete analyser (AQ2; Seal Analytical, Mequon, WI).

### 2.7 Data analysis

For all variables, we calculated descriptive statistics (mean, standard error, and IQRs) and checked all variables and models for normality of residuals and homoscedasticity. We transformed data using a log function for all variables. Within each site,

we used repeated measures ANOVA models to test for differences in $N_2O$ flux (g $N_2O$ N ha$^{-1}$ day$^{-1}$) across treatments for all time points. Models included day as the repeated measure, cover crop treatment as the fixed effect, and block as the random effect. We estimated mean cumulative $N_2O$ emissions (g $N_2O$ N ha$^{-1}$) for all treatments by calculating the area under the curve (Gelfand et al., 2016) using the following Eq. (2):

$$Cumulative\ N_2O\ Emissions = \sum_{t_0}^{t_{final}}[(x_t + x_{t+1})/2] * [(t+1) - t],\qquad(2)$$

Where $t_0$ is the initial sampling date, $t_{final}$ is the final sampling date, $x_t$ is $N_2O$ flux at time t, and $x_{t+1}$ is $N_2O$ flux at the following sampling date. In the absence of continuous sampling, this approach allowed us to approximate a total flux over the sampling window and better visualize treatment patterns within and across sites.

Within each site, we determined the effects of cover crop treatments on cumulative $N_2O$, total biomass (kg ha$^{-1}$), total biomass N (kg N ha$^{-1}$), shoot C:N ratio, clover N (kg N ha$^{-1}$), BNF (kg N ha$^{-1}$), and soil inorganic N using separate ANOVA models for a randomized complete block design, with cover crop treatment as the fixed effect and block as the random effect. To understand the effects of cover crop treatments on all response variables across both sites, we used two-way ANOVA models with site and treatment as fixed effects, along with their interaction, and block nested in site as a random effect. We tested for differences in soil inorganic N concentrations by site for each treatment between sampling dates using a t-test. For all ANOVAs, post-hoc comparison of least square means was performed using Tukey's HSD, and results were reported as statistically significant at $\alpha = 0.05$. For models including $N_2O$ fluxes we used $\alpha = 0.1$, following previous work identifying high variability from unidentified sources in ecological field experiments measuring $N_2O$ emissions (Gelfand et al., 2016; Han et al., 2017). All statistical analyses were performed in JMP Pro 15 software (SAS Institute, Cary NC). Excel and JMP Pro 15 were used to make figures.

## 3 Results

### 3.1 Soil Fertility

The *CF* site had higher soil fertility compared to the *KBS* site (Table 1). Total organic C was 34% higher at *CF* ($P = 0.0003$). Similarly, we found that *CF* had significantly larger POM pools than *KBS.* The concentration of free particulate organic matter (fPOM) was 44% higher ($P = 0.011$) and occluded particulate organic matter (oPOM) was 29% higher at *CF* ($P = 0.006$). The fPOM N concentration was 30% higher at *CF* than *KBS* ($P = 0.041$) and PMN was 46% higher at *CF* than at *KBS* ($P = 0.004$). However, oPOM N was not significantly different between *CF* and *KBS* ($P = 0.295$). Soil inorganic N increased during the $N_2O$ sampling period in all treatments at both sites. We found a significantly larger inorganic N pool at *CF* than *KBS* at both sampling dates ($P < 0.001$) (Table 2).

### 3.2 Cover crop biomass and traits (C:N and BNF)

There was a significant effect of site ($P = 0.0005$), treatment ($P < 0.0001$) and a significant interaction between site and treatment ($P = 0.008$) for total shoot biomass, which included both cover crops and weed species. Across all cover crop treatments, mean biomass was 40% higher at *CF* ($5430 \pm 499$ kg ha$^{-1}$) than at *KBS* ($3260 \pm 289$ kg ha$^{-1}$), with nearly three times more rye biomass and 1.5 times more mixture biomass at *CF* than *KBS*. At *CF*, rye biomass ($7709 \pm 387$ kg ha$^{-1}$) was 37% higher than biomass in the clover treatment ($4846 \pm 477$ kg ha$^{-1}$), and almost threefold higher than in the fallow ($2775 \pm 245$ kg ha$^{-1}$) ($P < 0.0001$). Rye and mixture ($6392 \pm 206$ kg ha$^{-1}$) were not significantly different from each other, nor were the mixture and clover treatments. At *KBS,* clover ($3972 \pm 580$ kg ha$^{-1}$) and mixture ($4219 \pm 297$ kg ha$^{-1}$) treatments had approximately twofold more biomass than the fallow ($2006 \pm 388$ kg ha$^{-1}$) ($P = 0.007$). However, mixture and clover biomass did not differ significantly from rye ($2842 \pm 212$ kg ha$^{-1}$), and rye was not significantly different from fallow (Figure 1). At both sites, clover performed well in the mixture, representing 54% of the total mixture biomass at *KBS* and 53% of total mixture biomass at *CF* (Table A1).

We also found a significant effect of site ($P = 0.0005$), treatment ($P < 0.0001$), and a significant site by treatment interaction ($P = 0.048$) on total shoot N (including both cover crop and weed biomass). Across sites, there was two-fold higher biomass N at *CF* ($102.6 \pm 8.7$ kg N ha$^{-1}$) than at *KBS* ($53.0 \pm 7.2$ kg N ha$^{-1}$), with 68% higher N in rye biomass, 44% higher in mixture, and 56% higher in fallow at *CF* compared to *KBS*. At *CF*, there was a significant difference in biomass N between treatments, in which clover ($121.2 \pm 14.4$ kg N ha$^{-1}$) accumulated twofold more N than the weeds in the fallow ($59.0 \pm 14.4$ kg N ha$^{-1}$) ($P = 0.006$); however, clover, mixture ($131.3 \pm 14.3$ kg N ha$^{-1}$), and rye ($98.6 \pm 4.6$ kg N ha$^{-1}$) treatments did not significantly differ from each other. At *KBS,* we found significantly higher aboveground N in the clover ($80.8 \pm 13.5$ kg N ha$^{-1}$) and mixture ($73.4 \pm 5.8$ kg N ha$^{-1}$) treatments compared to the rye ($31.9 \pm 1.4$ kg N ha$^{-1}$) and weedy fallow ($26.0 \pm 6.6$ kg N ha$^{-1}$) ($P = 0.0004$) (Figure 1).

There was also a significant effect of site ($P = 0.001$), treatment ($P < 0.0001$), and a significant interaction between site and treatment ($P = 0.005$) for cover crop C:N. Across sites for all treatments combined, C:N was 26% higher at *KBS* ($30.7 \pm 2.0$) than *CF* ($23.7 \pm 1.8$). At *CF*, the C:N of rye biomass was $34.7 \pm 1.6$, while the mixture had a significantly lower C:N ($21.7 \pm 1.8$). The mixture C:N did not differ from that in clover ($17.2 \pm 0.7$) or weeds in the fallow ($21.1 \pm 1.6$; $P < 0.0001$). At *KBS*, we also found a lower C:N in treatments with legumes ($40.3 \pm 1.3$ in rye and $34.8 \pm 1.9$ in fallow vs. $25.6 \pm 1.1$ in the mixture and $21.8 \pm 0.3$ in clover; $P < 0.0001$). At *KBS*, the difference between clover and mixture was not significant.

Using stable isotope methods at KBS, we estimated that the clover shoot N derived from fixation was 43.3 % when grown alone and 63.3 % when grown in mixture with rye, which we applied to estimates of N supply from BNF at both sites. There

was a weakly significant effect of site ($P = 0.053$) on N supplied by BNF in clover, but no significant effect of treatment ($P = 0.704$) and no significant interaction ($P = 0.936$). Between sites, with mixture and clover treatments combined, aboveground N from BNF was 38 % higher at $CF$ ($49.5 \pm 7.3$ kg N ha$^{-1}$) than at $KBS$ ($30.6 \pm 3.5$ kg N ha$^{-1}$) ($P = 0.053$). At $KBS$, BNF in clover ($29.2 \pm 6.0$ kg N ha$^{-1}$) and mixture ($32.1 \pm 4.4$ kg N ha$^{-1}$) were not significantly different ($P = 0.677$). Similarly, at $CF$, clover ($46.2 \pm 8.3$ kg N ha$^{-1}$) and mixture ($52.7 \pm 13.1$ kg N ha$^{-1}$) supplied similar BNF inputs ($P = 0.865$). In a sensitivity analysis for BNF at $CF$ spanning 40-70 %Ndfa, N from fixation ranged from 42.7 to 74.7 kg N ha$^{-1}$ for the sole clover treatment and from 33.3 to 58.3 kg N ha$^{-1}$ for the clover in the mixture treatment (Table A3).

### 3.3 Effects of a legume-grass cover crop mixture on daily N$_2$O emissions

In the repeated measures model for daily N$_2$O flux at $CF$, we found a significant effect of cover crop treatment ($P = 0.070$), day ($P < 0.0001$), and a significant interaction between day and treatment ($P = 0.005$). At $KBS$, there was a significant effect of cover crop treatment ($P = 0.016$) and day ($P < 0.0001$). Individual ANOVA models for each sampling date at $CF$ showed that N$_2$O emissions were higher in the clover ($4.5 \pm 0.5$ g N$_2$O N ha$^{-1}$), mixture ($4.8 \pm 1.3$ g N$_2$O N ha$^{-1}$), and rye ($7.7 \pm 2.2$ g N$_2$O N ha$^{-1}$) treatments than in the fallow ($1.2 \pm 0.3$ g N$_2$O N ha$^{-1}$) at the baseline sampling point prior to tillage ($P = 0.002$). Six days after incorporating the cover crops by tillage, N$_2$O emissions in the clover treatment peaked at $55.1 \pm 16.4$ g N$_2$O N ha$^{-1}$, whereas fluxes in the other treatments had started to decline (Figure 2 A). On day six, emissions in the clover treatment were significantly higher than in the fallow ($16.8 \pm 6.2$ g N$_2$O N ha$^{-1}$) ($P = 0.032$), whereas the mixture ($21.0 \pm 3.5$ g N$_2$O N ha$^{-1}$) and rye ($16.5 \pm 2.2$ g N$_2$O N ha$^{-1}$) treatments were not different from fallow. Emissions in the clover treatment remained elevated for the rest of the measurement period, however, the difference in emissions between clover, mixture, and rye treatments was not statistically significant on the last sampling date, 18 days after tillage ($P = 0.151$) (Figure 2 A).

At $KBS$, N$_2$O emissions were five times higher in the mixture ($18.0 \pm 5.6$ g N$_2$O N ha$^{-1}$) than in rye ($3.6 \pm 1.0$ g N$_2$O N ha$^{-1}$) at the peak flux eight days after tillage ($P = 0.049$) and were also five times higher in mixture ($9.4 \pm 2.6$ g N$_2$O N ha$^{-1}$) than the rye ($1.8 \pm 0.4$ g N$_2$O N ha$^{-1}$) eleven days after tillage ($P = 0.018$). Twelve days after tillage, emissions were four times higher in clover ($5.9 \pm 1.1$ g N$_2$O N ha$^{-1}$) than rye ($1.5 \pm 0.6$ g N$_2$O N ha$^{-1}$) ($P = 0.018$). By the fifteenth and last day, clover ($4.4 \pm 1.3$ g N$_2$O N ha$^{-1}$) and mixture ($7.2 \pm 1.6$ g N$_2$O N ha$^{-1}$) were higher than rye ($1.9 \pm 0.4$ g N$_2$O N ha$^{-1}$) and fallow ($1.7 \pm 0.3$ g N$_2$O N ha$^{-1}$) ($P = 0.007$) (Figure 2 B).

### 3.4 Cumulative N$_2$O emissions

Both cover crop treatment ($P = 0.002$) and site ($P = 0.004$) had a significant effect on cumulative N$_2$O emissions, with no significant interaction ($P = 0.138$). The mean N$_2$O flux following tillage was 1.8 times higher at $CF$ ($413.4 \pm 67.5$ g N$_2$O-N ha$^{-1}$ vs. $230.8 \pm 42.5$ g N$_2$O-N ha$^{-1}$; $P = 0.004$), which had both higher rates of potentially mineralizable N and larger free and occluded POM fractions (Figure 3). On average across both sites, the clover ($488.5 \pm 129.4$ g N$_2$O-N ha$^{-1}$) and mixture ($388 \pm$

46.2 g $N_2O$-N $ha^{-1}$) treatments led to significantly higher emissions than the rye (193.0 ± 43.4 g $N_2O$-N $ha^{-1}$) and fallow (218.0 ± 52.5 g $N_2O$-N $ha^{-1}$), with clover producing more than 2.5 times and mixture 2 times higher emissions than rye ($P = 0.002$). Emissions from clover and mixture were statistically similar, and emissions from rye and fallow also did not differ significantly.

When evaluating treatment effects within each site, at *CF*, cumulative $N_2O$ flux tended to be lower in the fallow (291.5 ± 92.0 g $N_2O$-N $ha^{-1}$), rye (288.9 ± 48.1 g $N_2O$-N $ha^{-1}$), and clover-rye mixture (380.2 ± 44.4 g $N_2O$-N $ha^{-1}$) treatments compared to clover grown alone (692.9 ± 204.7 g $N_2O$-N $ha^{-1}$), although these differences were not statistically significant ($P = 0.112$). At *KBS*, cumulative $N_2O$ fluxes were lower in the fallow (144.5 ± 28.2 g $N_2O$-N $ha^{-1}$) and rye (97.1 ± 18.3 g $N_2O$-N $ha^{-1}$) treatments compared to the clover-rye mixture (397.7 ± 89.1 g $N_2O$-N $ha^{-1}$) and clover grown alone (284.1 ± 91.5 g $N_2O$-N $ha^{-1}$) ($P = 0.008$). At this site, the mixture produced four times, and clover three times, higher emissions than rye (Figure 4).

### 3.5 $N_2O$ fluxes normalized by soil fertility indicators or cover crop biomass

Given the contrasting soil fertility properties at the two experimental sites, we normalized $N_2O$ emissions by POM levels and PMN rates (i.e., cumulative $N_2O$ to POM, or PMN, ratios). When controlling for differences in soil fertility, all ratios had significant treatment effects, with clover resulting in the highest $N_2O$ emissions at *CF* and mixture producing the highest emissions at *KBS* (Table 3). There was no significant effect of site on cumulative $N_2O$ when expressed per unit fPOM or PMN. However, when normalizing for differences in oPOM, oPOM N, and fPOM N across sites, there was a significant site effect. Specifically, compared to *KBS*, mean $N_2O$ emissions at *CF* were 22% higher when normalizing for oPOM ($P = 0.011$), 43% higher for oPOM N ($P = 0.001$), and 26% higher for fPOM N ($P = 0.027$). When normalized by POM fractions or PMN, the cumulative $N_2O$ emissions across sites were 1.9-2.8 times higher in clover and mixture than in fallow or rye (Table 4). When $N_2O$ was normalized by cover crop biomass, site was not significant ($P = 0.180$), but we found a significant treatment effect ($P = 0.003$) with lower emissions following rye than the other treatments. There was no effect of either treatment ($P = 0.171$) or site ($P = 0.467$) when expressing $N_2O$ emissions as a ratio of cover crop biomass N (Table 5). Daily $N_2O$ fluxes normalized by cover crop biomass and biomass N are presented in the Appendix (Table A2).

### 4 Discussion

Reducing greenhouse gas emissions from agriculture is necessary to meet global targets for limiting climate change (IPCC, 2019). Generally, greenhouse gas emissions are greater from grain agroecosystems with fertilizer additions compared to legume N sources (Robertson et al., 2014; Han et al., 2017; Westphal et al., 2018) and are higher in rotations with only annual crops compared to those with perennial crops (Gelfand et al., 2016). Overwintering cover crops can help "perennialize" annual agroecosystems by providing continuous plant cover, building SOC (King and Blesh, 2018) and supporting related functions

such as soil nutrient supply and storage. In diversified rotations with cover crops, however, $N_2O$ emissions can peak during the weeks following tillage when cover crop biomass is incorporated into the soil, increasing N mineralization rates (Han et al., 2017). There is growing evidence that small increases in cover crop functional diversity can simultaneously enhance multiple agroecosystem functions, including nutrient retention (Storkey et al., 2015; Blesh, 2017; Kaye et al., 2019). For instance, Storkey et al. (2015) found that low to intermediate levels of species richness (1-4 species) provided an optimal balance of multiple ecosystem services when species exhibited contrasting functional traits related to growth habit and phenology. Our experiment tested whether increasing cover crop functional diversity with a legume-grass mixture, compared to a sole legume, would reduce pulse $N_2O$ emissions following cover crop incorporation by tillage at two field sites. Understanding these critical moments of $N_2O$ flux can inform how to adapt management of diversified cropping systems to reduce N losses, and further reap their environmental benefits compared to fertilizer-based management practices.

## 4.1 Effects of a legume-grass cover crop mixture on $N_2O$ flux

The sampling period (15-18 days) of this experiment captured the first peak of $N_2O$ emissions following tillage of cover crop biomass at both sites. Our analysis of cover crop treatment effects on cumulative $N_2O$ emissions in this period shows the strong influence of biomass N inputs, particularly for the legume species, which supplied an external N source through BNF. When normalized for differences in soil fertility across sites, the clover and mixture treatments led to significantly higher pulse losses of $N_2O$ than rye or fallow (Table 4), providing strong evidence that BNF inputs from the treatments that included clover were a driving factor of $N_2O$ losses. While our study tested the role of legume N inputs, prior research, summarized in recent meta-analyses, has been dominated by studies with synthetic fertilizer and manure N sources (Han et al., 2017; Eagle et al., 2017; Basche et al., 2014). The only studies included in these meta-analyses that had legumes as the sole N source were Robertson et al. (2000) and Alluvione et al. (2010), both using tillage to terminate cover crops. Gelfand et al. (2016) extended the data reported in the Robertson et al. (2000) study by another decade and found that legume N sources did not significantly reduce $N_2O$ fluxes from soil compared to fertilizer N sources. Our findings contribute evidence that legume cover crops release more $N_2O$ compared to treatments without legumes, within the context of agroecosystems that have only received legume N inputs for several decades.

Despite clear differences between treatments with clover and those without, we did not find strong support for our hypothesis that the legume-grass mixture would reduce pulse $N_2O$ flux. This may be explained by the lack of difference in total BNF inputs between clover grown alone and in mixture within each site, as well as the similar C:N ratios of litter biomass in both treatments. Litter chemistry for clover and mixture both fell into the intermediate C:N range (17.2-25.6) expected to lead to net N mineralization compared to the much higher C:N in rye (31.5-44.1) across sites, which likely led to net N immobilization (Robertson and Groffman, 2015; Kramberger et al., 2009; Rosecrance et al., 2000). Indeed, the soil inorganic N concentration, which exerts a direct control on $N_2O$ flux (Robertson et al. 2000), increased at both sites over the sampling period, and was significantly higher in clover compared to rye, while clover and mixture were not different.

When $N_2O$ fluxes were normalized by aboveground biomass N, emissions were the same for all treatments regardless of the source of N (internal cycling of soil N or external inputs of fixed $N_2$). Furthermore, rye biomass N, which was three-fold higher at *CF* than at *KBS*, corresponded with 1.6-2.6 times higher $N_2O$ emissions at *CF* when normalized to control for differences in soil fertility across sites. BNF inputs in the clover treatment were 1.5 times higher at *CF,* which corresponded with 1.2-2.3 times higher $N_2O$ emissions when normalized by soil fertility properties. Greater clover biomass in both treatments with clover at *CF* corresponded with significantly higher BNF inputs and $N_2O$ emissions at that site. However, when $N_2O$ fluxes were normalized by aboveground biomass, emissions were significantly lower following rye than the other treatments, including weeds in the fallow, indicating that residue traits such as C:N influence $N_2O$ emissions. Higher mean litter C:N in the rye litter compared to the other treatments may have reduced $N_2O$ emissions per unit biomass input. These results reflect the importance of cover crop functional type, and the impact of legume N fixation inputs on episodic $N_2O$ emissions, which is supported by prior studies showing that higher total N inputs lead to higher N mineralization rates and higher $N_2O$ fluxes (e.g., Han et al., 2017) and that legume cover crops can lead to pulse $N_2O$ fluxes following incorporation by tillage (Baggs et al., 2003; Millar et al., 2004; Basche et al., 2014).

Within each site, the specific treatment effects differed. At *CF*, the clover treatment produced the highest pulse of $N_2O$, while at *KBS*, the mixture produced the highest flux, with the magnitude of the treatment effect being much more pronounced. $N_2O$ fluxes were four times higher following mixture than rye at *KBS*, compared to just over two times higher in clover than rye at *CF*, suggesting that the new N input from BNF was a stronger driver of treatment differences in the lower fertility soil (*KBS*). At *CF*, the mixture slightly reduced cumulative $N_2O$ emissions compared to clover (380.2 v. 692.9 g $N_2O$-N ha$^{-1}$), a difference which was likely ecologically meaningful even though it was not statistically significant. In contrast, at *KBS*, both treatments with clover produced significantly higher $N_2O$ emissions than the non-legume treatments.

In addition, differences between cover crop treatments may have been even greater at *CF* than our data suggests. We likely underestimated cumulative $N_2O$ emissions during the first peak following tillage at *CF* because emissions had not yet returned to baseline, especially for the clover treatment. By extending our empirical measurements using regression models, we estimated the trajectory of $N_2O$ emissions to approximately 19-26 days after tillage depending on the cover crop treatment and replicate. Cumulative $N_2O$ emissions at *CF* could have reached 822.8 ± 253.2 g $N_2O$ N ha$^{-1}$ in clover, 461.6 ± 59.2 g $N_2O$ N ha$^{-1}$ in mixture, 340.4 ± 63.4 g $N_2O$ N ha$^{-1}$ in rye, and 355.0 ± 77.4 g $N_2O$ N ha$^{-1}$ in fallow. These higher estimates also further increase differences in cumulative $N_2O$ emissions between sites.

At both sites, the clover was competitive in mixture, representing just over half of the total stand biomass in this treatment. Although similar mixture composition allowed for better comparison of this treatment between sites, there is a need for future

studies to assess a range of legume-to-grass ratios because mixture composition influences the quality of cover crop residue inputs to soil (Finney, White, and Kaye, 2016) and mixture evenness is related to agroecosystem multifunctionality (Blesh et al. 2019). For example, it is possible that if rye had produced more biomass in the mixture in our experiment, we would have observed lower $N_2O$ emissions in the mixture compared to the clover treatment at both sites.

## 4.2 Differences in $N_2O$ flux between sites

The different treatment patterns for daily emissions between sites, and the larger pulse emissions overall at *CF*, both provide insights into mechanisms governing $N_2O$ fluxes following cover crop incorporation. Although new N inputs to agroecosystems are a primary driver of soil $N_2O$ emissions (e.g., Han et al., 2017, Robertson and Groffman, 2015), in our study mean BNF inputs did not significantly differ between clover and mixture treatments. Thus, the different baseline soil fertility levels, and rhizosphere interactions that drive N mineralization, likely played a key role in the contrasting effects of the mixture across sites. For instance, prior studies have found positive correlations between total SOC and $N_2O$ flux (Bouwman et al., 2002; Dhadli et al., 2016) and Basche et al. (2014) found that both SOC and cover crop biomass had a significant effect on denitrification potential and $N_2O$ emissions. These studies highlight the important role of ecosystem state factors that influence fertility, such as soil parent material and organic C content, in driving $N_2O$ emissions.

Here, we found approximately twofold higher cumulative $N_2O$ fluxes at the site with larger soil POM fractions and higher POM N concentrations (*CF*) (Figure 3), suggesting that POM fractions influence cover crop growth and $N_2O$ fluxes. POM fractions are robust indicators of soil fertility that respond to changes in management over shorter time scales than total SOM and play an important functional role in soil N cycling and N availability to crops (Wander, 2004; Luce et al., 2016). For instance, the *CF* site also had approximately twofold higher rates of N mineralization (PMN) and 5 times higher soil inorganic N concentrations compared to *KBS*. The total amount of soil N assimilated by cover crops (in the absence of external N inputs) is also an integrated indicator of soil inorganic N availability over the cover crop season. Rye aboveground biomass N was threefold higher at *CF*, while N in weed biomass in the fallow control was 2.3 times higher at *CF* than at *KBS*. In diversified agroecosystems, plant-mediated N acquisition from SOM pools can couple the release of inorganic N with plant N uptake in the rhizosphere (Paterson et al., 2006), making organic N inputs, such as those from legume residues, less susceptible to loss than inorganic fertilizer inputs (Drinkwater and Snapp, 2007). Cover crops in higher fertility soils are thus likely to have higher net primary productivity, and to release more root C into the soil, which increases microbial growth and turnover rates, and mineralizes more soil N. The roots, in turn, take up more N and produce more biomass (Hodge et al., 2000; Paterson et al., 2006). This positive feedback loop may have led to the significantly higher cover crop biomass production at *CF*, which was especially pronounced in the rye treatment (7709 kg ha$^{-1}$ at *CF* compared to 2842 kg ha$^{-1}$ in at *KBS*).

Mechanistically, interactions between background soil fertility and cover crop functional types likely drive soil inorganic N availability and $N_2O$ emissions. For instance, the highest $N_2O$ emissions measured in our study were from the clover treatment

at *CF*, which had both the highest new N inputs to soil from BNF and the largest POM pools. This site also showed a small reduction in emissions with the legume-grass mixture. After clover incorporation, the large, relatively labile C and N input to soil, in combination with larger background POM pools, may have primed greater overall N mineralization at *CF* compared to *KBS*, with some of this N lost as $N_2O$. Since corn had not yet established during this two-week period after tillage, there were no active roots to couple N release with N uptake, allowing soil inorganic N pools to increase (Table 2) and leaving a window of opportunity for N losses.

Even when controlling for fertility differences across sites (i.e., the analysis of $N_2O$ to POM or PMN ratios), we found that cumulative $N_2O$ emissions per unit oPOM, oPOM N, and fPOM N were significantly higher at *CF*. This site difference was highest for the oPOM N stock, with about 43% more emissions per oPOM N at *CF*. Prior studies have shown that oPOM N is a strong indicator of SOM quality, N fertility, and soil inorganic N availability from microbial turnover of SOM (Marriott and Wander, 2006; Bu et al., 2015; Blesh, 2019). Our contrasting findings across experimental sites indicate a need for future studies that assess the effects of cover crops on $N_2O$ emissions across soils with a wide range of POM pool sizes.

**4.3 Episodic $N_2O$ emissions following tillage of cover crops**

To understand the relative importance of $N_2O$ fluxes following cover crop incorporation, it is important to interpret the magnitude of these episodic emissions within the context of $N_2O$ fluxes for a complete crop rotation. In a 20-year study in the biologically-based cropping system in the MCSE at KBS (the *KBS* site in our experiment), Gelfand et al. (2016) reported mean annual $N_2O$ emissions of approximately 1.08 kg N ha$^{-1}$ yr$^{-1}$ during a corn year, which was defined as the 380-day window between corn planting and soybean planting the following year. They also calculated an average of 2.2 kg N ha$^{-1}$ yr$^{-1}$ over the course of the three-year corn-soy-wheat crop rotation at this site (Gelfand et al., 2016). These values are likely a slight underestimate because their sampling did not include emissions during winter thaws, and occurred every 2 weeks, potentially missing periods of high emissions. In a meta-analysis, Han et al. (2017) reported a similar average annual $N_2O$ flux of 2.3 – 3.1 kg N ha$^{-1}$ yr$^{-1}$ for annual cropping systems with inorganic fertilizer additions.

Using Gelfand et al.'s estimate of 1.08 kg N ha$^{-1}$ yr$^{-1}$, the two-week cumulative flux we measured post-tillage of clover would represent 62.6% of crop year emissions at *CF* and 26.3% at *KBS*, while the flux following tillage of the mixture biomass would represent 33.9% of the crop year estimate at *CF* and 37.8% at *KBS*. Using the estimate of 2.2 kg N ha$^{-1}$ yr$^{-1}$ for the complete crop rotation, the two-week cumulative flux we measured post-tillage of clover would represent 30.7% of annual emissions at *CF* and 12.9% at *KBS*, while the flux following tillage of the mixture biomass is 16.7% of that annual estimate at *CF* and 18.1% at *KBS*. After incorporating sole clover biomass, the average daily flux was 37.6 g N ha$^{-1}$ d$^{-1}$ at *CF* and 18.9 g N ha$^{-1}$ d$^{-1}$ at *KBS*, and after mixture biomass, was 20.4 g N ha$^{-1}$ d$^{-1}$ at *CF* and 26.5 g N ha$^{-1}$ d$^{-1}$ at *KBS;* these rates are approximately three- to twelve-fold greater than the mean daily flux reported for the organic cropping system at *KBS* (Gelfand et al., 2016).

Taken together, these comparisons highlight the relative importance of episodic $N_2O$ emissions following tillage of cover crops.

Additionally, we used long-term measurements of $N_2O$ emissions from the biologically-based cropping system at KBS as further context for interpreting our single-season results. Between 2014 and 2020, following the red clover cover crop, there were three years in which $N_2O$ fluxes were measured roughly two weeks apart within a month after tillage. These two-week periods of $N_2O$ emissions after incorporating red clover represented $19.9 \pm 2.1$ % of the annual emissions from this cropping system (Robertson 2020). These $N_2O$ measurements from past years at the KBS site were not collected until at least 8 days

after tillage, and likely missed the initial flux immediately following soil disturbance, which may explain why we found a slightly higher proportion of annual emissions (26.3%) following clover incorporation at *KBS*. These historical data suggest that we indeed captured the peak $N_2O$ flux following soil disturbance by tillage in our one-year experiment. Sampling frequently during the days and weeks following tillage of cover crops is therefore important for advancing knowledge of episodic emissions.

**5 Conclusion**

We tested the impacts of cover crop functional type on short-term N cycling dynamics following tillage in the context of diversified agroecosystems that rely on legume N. Given that gaseous N fluxes are episodic, it is critical to understand how they are influenced by management practices during periods of high susceptibility for N losses. Overall, $N_2O$ flux was higher in the clover and mixture treatments than in rye and fallow when emissions were normalized by soil fertility properties. We

found that the legume-grass cover crop led to a small reduction in $N_2O$ losses at *CF* but not at *KBS*. In contrast to our hypothesis, at *KBS*, the mixture led to higher $N_2O$ emissions than the clover treatment at peak flux following tillage. We also found a more pronounced treatment effect at *KBS*, indicating that new N inputs from both treatments with legumes were a larger driver of $N_2O$ emissions at the site with lower soil fertility. Overall, the clover treatment at *CF* led to the highest emissions across sites, suggesting a synergistic effect of BNF inputs and soil fertility on $N_2O$. These contrasting findings across sites shed light on

the drivers of $N_2O$ losses following cover crop incorporation. Our results show that higher aboveground cover crop biomass can lead to higher $N_2O$ emissions during cover crop decomposition, particularly for cover crops that include legumes.

**Data availability:** Data is available in Deep Blue Repositories at https://doi.org/10.7302/hv7v-4378

**Author contribution:** AB and JB developed the research questions, experimental design, and methods. AB conducted the

470 field and lab work and led data analysis with input from JB. AB and JB wrote and edited the manuscript.

**Competing interests:** The authors declare that they have no conflict of interest.

**Acknowledgments:** We thank Brendan O'Neill and Kevin Kahmark for assistance with the $N_2O$ sampling protocol, Jeremy Moghtader and Joe Simmons for managing field operations, and Beth VanDusen for technical support in the field and the lab.

We would also like to thank Dev Gordin, Kent Connell, Marta Plumhoff, Etienne Herrick, Luyao Li, Nicole Rhoads, Kristen Hayden, Danielle Falling, Riley Noble, Alice Elliott, Ben Iuliano, Dahlia Rockowitz, Ellie Katz, Naveen Jasti, and Nisha Gudal for assistance in the field and lab, and Don Zak for input on a draft of the manuscript. Support for this research was also provided by the NSF Long-term Ecological Research Program (DEB 1832042) at the Kellogg Biological Station and by Michigan State University AgBioResearch.

**Financial support**: This work was supported by Research Funding for Conservation Studies from the Matthaei Botanical Gardens, University of Michigan, a Rackham Graduate Student Research Grant from the University of Michigan, and a United States Department of Agriculture (USDA) NIFA grant (Award #: 2019-67019-29460).

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

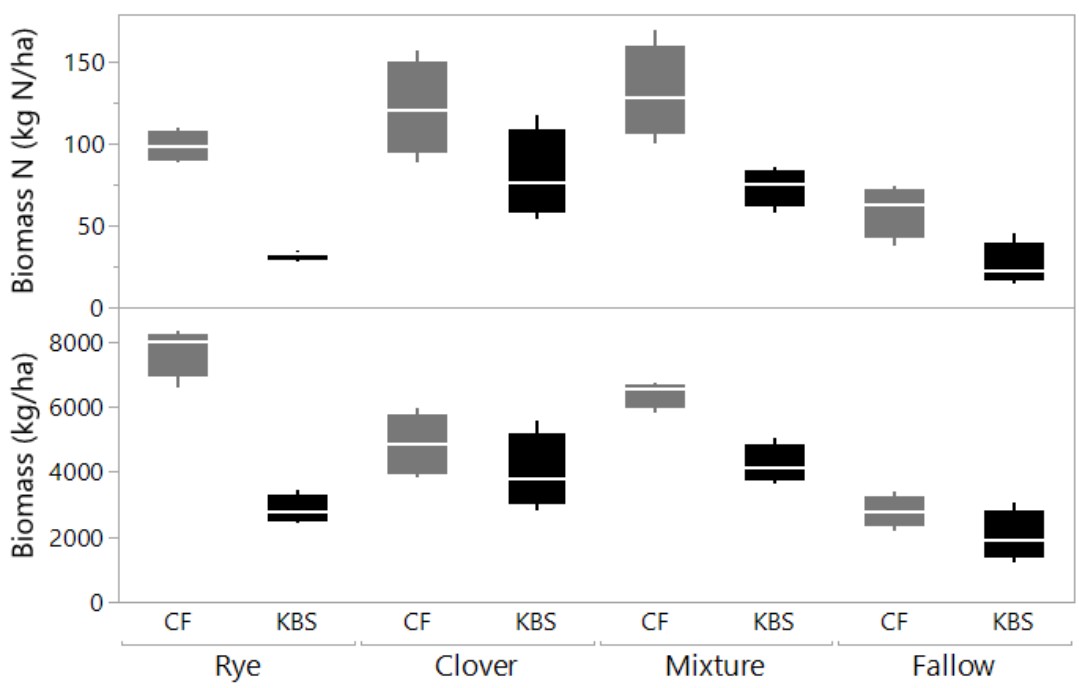

**Figure 1: Biomass (kg/ha) and biomass N (kg N/ha) by treatment (including cover crops and weeds), at two sites (*CF* and *KBS*).**

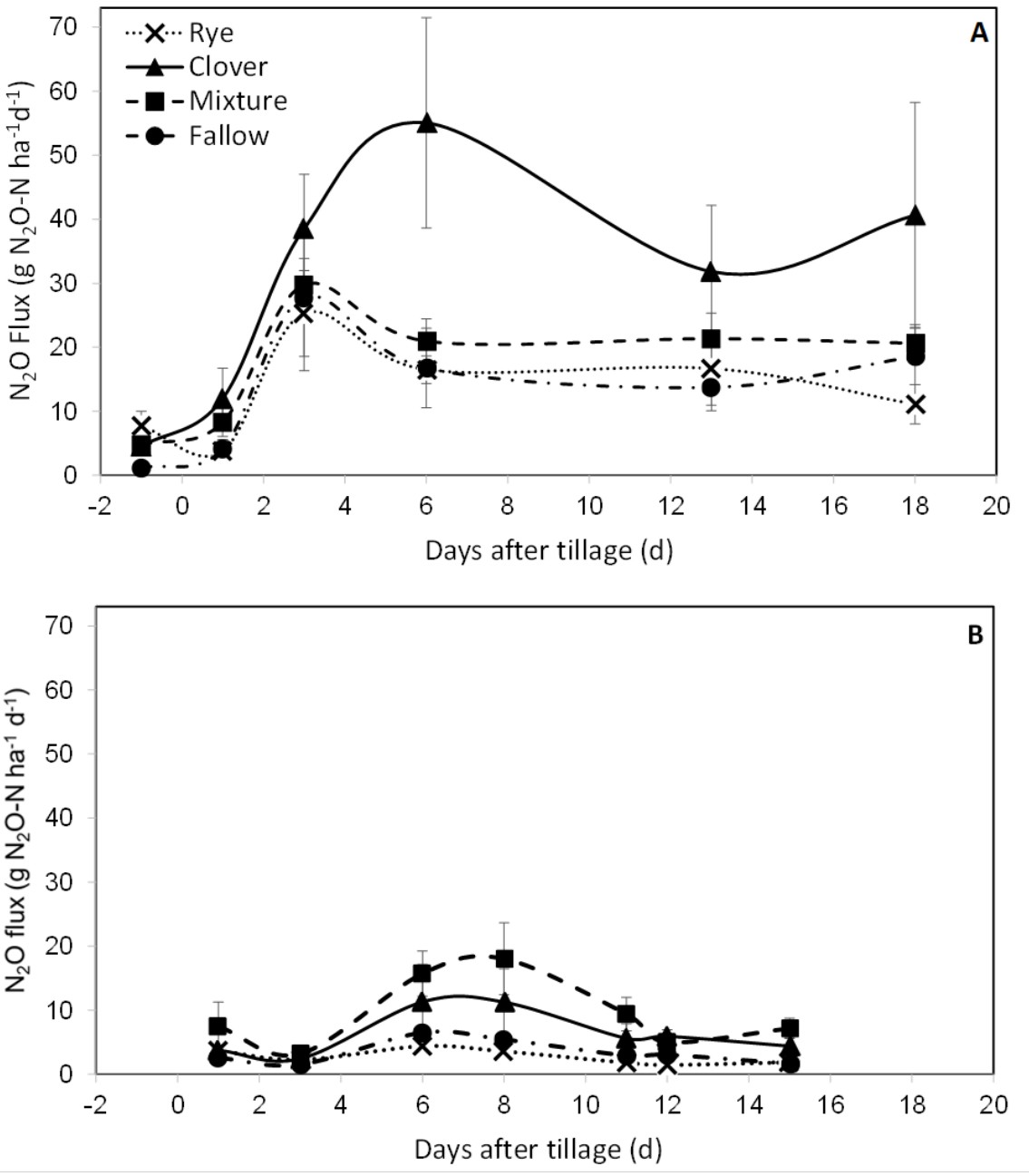

**Figure 2: A:** Mean net nitrous oxide ($N_2O$) flux from the soil (with standard error) over 18 days at *CF*, measured once the day before (d = -1) tillage on 23 May 2018 (d = 0), and then five times following tillage and incorporation of cover crop biomass. **B:** Mean net nitrous oxide ($N_2O$) flux from the soil (with standard error) over 15 days at *KBS*, measured seven times following tillage on 26 May 2020 (d = 0). The lines connecting the sampling points are intended to aid in visualizing treatment patterns for cumulative $N_2O$ and do not indicate continuous data collection (Eq. 2).

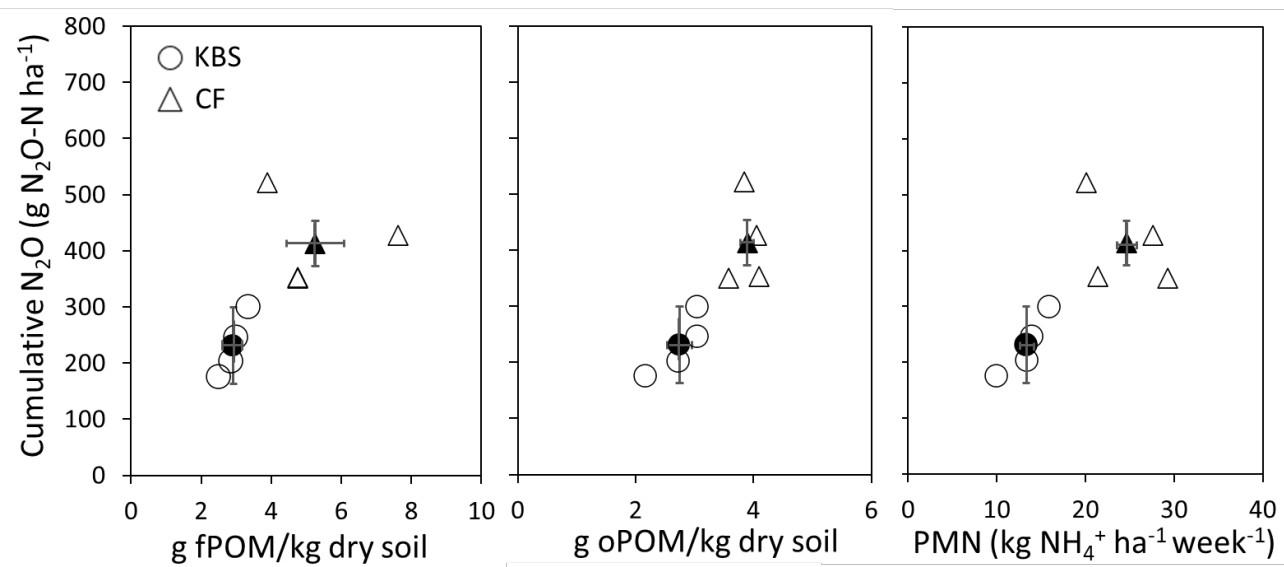

**Figure 3: Cumulative N₂O plotted against fPOM (g kg⁻¹), oPOM (g kg⁻¹), and PMN (kg NH₄⁺ N ha⁻¹ week⁻¹) at both sites (*KBS* and *CF*). Open symbols are values by replicate block and closed symbols are overall site means. Error bars represent standard error of the means for each site.**

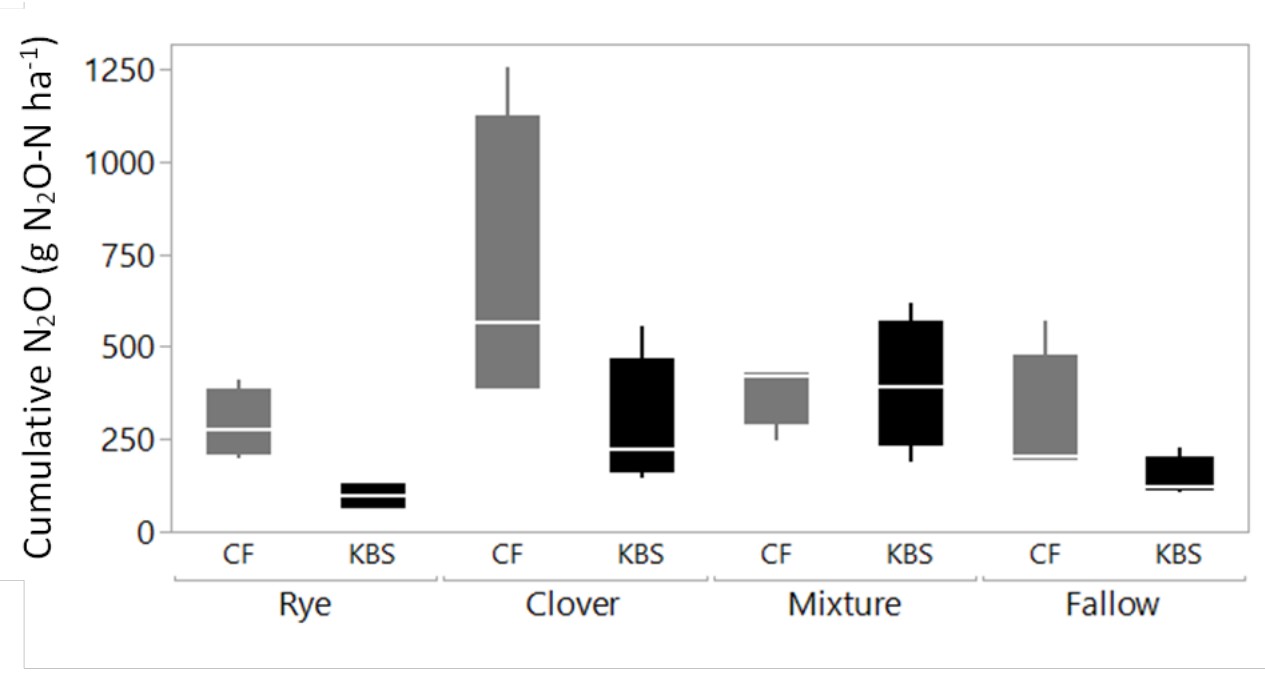

**Figure 4: Cumulative N₂O flux by treatment, compared between sites.**

**Table 1: Soil fertility indicators at each site. P-values are indicated as: * <0.05, ** <0.001 for differences between sites.**

| | CF | | KBS | |
| --- | --- | --- | --- | --- |
| Soil Series | Fox | | Kalamazoo & Oshtemo | |
| | Mean | Std. Error | Mean | Std. Error |
| *Bulk Density | 1.48 | 0.02 | 1.58 | 0.02 |
| **% Sand | 65.00 | 1.29 | 41.30 | 2.06 |
| % Clay | 21.50 | 0.96 | 19.40 | 1.33 |
| **% Silt | 13.50 | 0.50 | 39.30 | 2.40 |
| pH | 6.35 | 0.20 | 6.59 | 0.07 |
| **Total Organic Carbon (Mg ha$^{-1}$) | 44.39 | 1.81 | 29.44 | 1.01 |
| **Total Organic Nitrogen (Mg ha$^{-1}$) | 3.83 | 0.10 | 2.81 | 0.06 |
| *Phosphorus (mg P kg$^{-1}$) | 16.00 | 1.91 | 9.31 | 1.85 |
| Potassium (mg K kg$^{-1}$) | 62.25 | 5.31 | 60.19 | 3.18 |
| *oPOM (mg kg$^{-1}$) | 3.89 | 0.05 | 2.75 | 0.14 |
| oPOM N (mg N kg$^{-1}$) | 63.20 | 1.05 | 56.93 | 2.95 |
| *fPOM (mg kg$^{-1}$) | 5.26 | 0.36 | 2.92 | 0.13 |
| *fPOM N (mg N kg$^{-1}$) | 62.31 | 3.69 | 43.54 | 2.11 |
| *PMN (kg NH$_4^+$ N ha$^{-1}$ week$^{-1}$) | 24.62 | 1.01 | 13.34 | 0.90 |


**Table 2: Mean ± standard error for soil inorganic N (mg N kg soil$^{-1}$) at initial and final sampling points separated by site and treatment. There was a significant difference between sites at both time points (<0.0001). P-values are indicated as: * <0.05, ** <0.01, *** <0.001 for differences between time points for each treatment in the last column. Significant treatment differences (within each**

**site) are indicated by different letters.**

| Soil Inorganic N | | | | |
| --- | --- | --- | --- | --- |
| Site | Treatment | Initial | Final | P-value |
| CF | Rye | $5.0 \pm 0.6\ b$ | $12.7 \pm 1.3\ b$ | * |
| | Clover | $9.0 \pm 0.6\ a$ | $26.7 \pm 1.6\ a$ | ** |
| | Clover-Rye | $7.1 \pm 0.7\ a$ | $20.0 \pm 1.9\ ab$ | ** |
| | Fallow | $4.5 \pm 0.2\ b$ | $17.5 \pm 4.3\ ab$ | * |
| KBS | Rye | $1.2 \pm 0.1\ bc$ | $2.8 \pm 0.5\ b$ | *** |
| | Clover | $1.7 \pm 0.2\ a$ | $5.0 \pm 0.8\ a$ | *** |
| | Clover-Rye | $1.5 \pm 0.2\ ab$ | $4.6 \pm 0.4\ a$ | *** |
| | Fallow | $0.9 \pm 0.2\ c$ | $3.3 \pm 0.5\ b$ | * |

**Table 3: Mean ± standard error for ratios of g $N_2O$/g POM and g $N_2O$/ kg PMN by treatment and site. P-values are indicated as: \* <0.05, \*\* <0.001 for differences between treatments, and ^ <0.05 for differences between sites.**

| Site | Treatment | $N_2O$/ fPOM* | $N_2O$/ oPOM*^ | $N_2O$/ fPOM N*^ | $N_2O$/ oPOM N*^ | $N_2O$/ PMN** |
|------|-----------|---------------|----------------|------------------|------------------|----------------|
| CF | Rye | 0.19 ± 0.03 | 0.25 ± 0.04 | 16.12 ± 3.08 | 15.36 ± 2.35 | 12.09 ± 2.48 |
| | Clover | 0.51 ± 0.19 | 0.60 ± 0.18 | 41.44 ± 14.96 | 37.82 ± 11.77 | 29.95 ± 11.04 |
| | Clover-Rye | 0.26 ± 0.04 | 0.33 ± 0.03 | 21.38 ± 3.31 | 20.27 ± 2.15 | 16.17 ± 2.84 |
| | Fallow | 0.17 ± 0.03 | 0.25 ± 0.08 | 14.94 ± 2.53 | 15.26 ± 4.29 | 11.67 ± 3.06 |
| KBS | Rye | 0.10 ± 0.02 | 0.13 ± 0.02 | 6.65 ± 1.38 | 5.82 ± 0.82 | 7.43 ± 1.14 |
| | Clover | 0.30 ± 0.09 | 0.34 ± 0.10 | 19.81 ± 6.54 | 15.80 ± 4.66 | 23.61 ± 6.49 |
| | Clover-Rye | 0.50 ± 0.12 | 0.47 ± 0.11 | 32.64 ± 8.50 | 22.00 ± 5.44 | 33.41 ± 7.85 |
| | Fallow | 0.16 ± 0.03 | 0.15 ± 0.03 | 10.50 ± 1.97 | 7.00 ± 1.39 | 9.33 ± 1.55 |

**Table 4: Mean ± standard error for ratios of g $N_2O$/g POM and g $N_2O$/ kg PMN averaged across both sites by treatment. Significant treatment differences are indicated by different letters.**

| Treatment | $N_2O$/ fPOM | $N_2O$/ oPOM | $N_2O$/ fPOM N | $N_2O$/ oPOM N | $N_2O$/ PMN |
|-----------|--------------|--------------|----------------|----------------|-------------|
| Rye | 0.15 ± 0.03*b* | 0.19 ± 0.03*b* | 11.39 ± 2.37*b* | 10.59 ± 2.14*b* | 9.76 ± 1.54*b* |
| Clover | 0.40 ± 0.11*a* | 0.47 ± 0.11*a* | 30.63 ± 8.59*a* | 26.81 ± 7.19*a* | 26.78 ± 6.05*a* |
| Clover-Rye | 0.38 ± 0.08*a* | 0.40 ± 0.06*a* | 27.01 ± 4.73*a* | 21.13 ± 2.73*a* | 24.79 ± 5.05*a* |
| Fallow | 0.17 ± 0.02*b* | 0.20 ± 0.04*b* | 12.72 ± 1.71*ab* | 11.13 ± 2.61*b* | 10.50 ± 1.65*b* |

**Table 5: Mean ± standard error for ratios of g $N_2O$ to kg cover crop biomass and g $N_2O$ to kg cover crop biomass N averaged across both sites by treatment. Significant treatment differences are indicated by different letters.**

| Treatment | $N_2O$/biomass | $N_2O$/biomass N |
|-----------|----------------|------------------|
| Rye | 0.036 ± 0.0049*b* | 2.98 ± 0.34*a* |
| Clover | 0.12 ± 0.034*a* | 5.12 ± 1.48*a* |
| Clover-Rye | 0.076 ± 0.011*a* | 4.17 ± 0.70*a* |
| Fallow | 0.087 ± 0.012*a* | 5.37 ± 0.60*a* |

## Appendix A

**Table A1: Means (standard error) for aboveground biomass, biomass nitrogen, and biological nitrogen fixation (BNF) by species across treatments at *CF* (A) and *KBS* (B).**

**A.**

| CF | All Cover Crops Biomass (kg ha$^{-1}$) | All Cover Crops Biomass N (kg N ha$^{-1}$) | Clover Biomass (kg ha$^{-1}$) | Clover Biomass N (kg N ha$^{-1}$) | Clover BNF (kg N ha$^{-1}$) | Rye Biomass (kg ha$^{-1}$) | Rye Biomass N (kg N ha$^{-1}$) | Weeds Biomass (kg ha$^{-1}$) | Weeds Biomass N (kg N ha$^{-1}$) |
|---|---|---|---|---|---|---|---|---|---|
| **Rye** | 7709.1 (387.2) | 98.6 (4.6) | | | | 7250.9 (341.7) | 89.2 (7.6) | 458.2 (201.3) | 9.4 (4.1) |
| **Clover** | 4845.8 (477.9) | 121.2 (14.4) | 4294.6 (680.5) | 106.7 (19.2) | 46.2 (8.3) | | | 551.2 (284.3) | 14.5 (6.5) |
| **Mixture** | 6392.4 (205.8) | 131.3 (14.4) | 3371.9 (702.6) | 83.3 (20.7) | 52.7 (13.1) | 2863.5 (495.4) | 43.9 (6.6) | 157.0 (70.4) | 4.1 (1.8) |
| **Fallow** | 2774.5 (245.1) | 59.0 (7.9) | | | | | | 2774.5 (245.1) | 59.0 (7.9) |

**B.**

| KBS | All Cover Crops Biomass (kg ha$^{-1}$) | All Cover Crops Biomass N (kg N ha$^{-1}$) | Clover Biomass (kg ha$^{-1}$) | Clover Biomass N (kg N ha$^{-1}$) | Clover BNF (kg N ha$^{-1}$) | Rye Biomass (kg ha$^{-1}$) | Rye Biomass N (kg N ha$^{-1}$) | Weeds Biomass (kg ha$^{-1}$) | Weeds Biomass N (kg N ha$^{-1}$) |
|---|---|---|---|---|---|---|---|---|---|
| **Rye** | 2842.8 (212.2) | 31.9 (1.4) | | | | 2367.7 (161.8) | 25.4 (0.5) | 475.2 (89.9) | 6.5 (1.1) |
| **Clover** | 3972.1 (579.7) | 80.8 (13.5) | 2963.9 (654.8) | 67.5 (14.0) | 29.2 (6.0) | | | 1008.2 (90.4) | 13.3 (1.2) |
| **Mixture** | 4219.1 (297.2) | 73.4 (5.8) | 2310.0 (380.7) | 50.6 (7.0) | 32.1 (4.4) | 1148.9 (300.9) | 13.1 (3.6) | 760.3 (43.3) | 9.6 (0.6) |
| **Fallow** | 2005.8 (387.9) | 26.0 (6.6) | | | | | | 2005.8 (387.9) | 26.0 (6.6) |

**Table A2: Means (standard error) for ratios of mg $N_2O$ to kg cover crop biomass (A) and for ratios of mg $N_2O$ to kg biomass N (B) by site and by treatment for each $N_2O$ sampling date**

**A. mg $N_2O$/kg cover crop biomass**

| Site | Treatment | 5/21/2018 | 5/23/2018 | 5/25/2018 | 5/28/2018 | 6/4/2018 | 6/9/2018 | |
|------|-----------|-----------|-----------|-----------|-----------|----------|----------|---|
| CF | Rye | 1.0 (0.3) | 0.5 (0.1) | 3.4 (1.1) | 2.2 (0.4) | 2.2 (0.8) | 1.4 (0.4) | |
| | Clover | 1.0 (0.1) | 2.6 (1.1) | 8.3 (2.1 | 12.5 (4.8) | 7.2 (3.1) | 9.5 (4.9) | |
| | Clover-Rye | 0.8 (0.2) | 1.3 (0.3) | 4.6 (0.5) | 3.3 (0.5) | 3.4 (0.7) | 3.3 (0.5) | |
| | Fallow | 0.4 (0.1) | 1.5 (0.5) | 9.6 (3.1) | 5.7 (1.6) | 4.8 (0.9) | 6.5 (1.0) | |
| | | 5/29/2020 | 5/31/2020 | 6/3/2020 | 6/5/2020 | 6/8/2020 | 6/9/2020 | 6/12/2020 |
| KBS | Rye | 1.3 (0.3) | 0.8 (0.4) | 1.5 (0.3) | 1.3 (0.4) | 0.6 (0.2) | 0.5 (0.2) | 0.6 (0.1) |
| | Clover | 1.1 (0.3) | 0.7 (0.3) | 3.1 (1.7) | 3.0 (1.5) | 1.5 (0.7) | 1.5 (0.3) | 1.1 (0.3) |
| | Clover-Rye | 1.8 (1.0) | 0.8 (0.2) | 3.6 (0.7) | 4.1 (1.2) | 2.2 (0.6) | 1.2 (0.3) | 1.7 (0.4) |
| | Fallow | 1.4 (0.4) | 0.9 (0.3) | 3.3 (0.1) | 2.9 (0.6) | 1.6 (0.3) | 1.3(0.4) | 0.9 (0.1) |

**B. mg $N_2O$/kg cover crop biomass N**

| Site | Treatment | 5/21/2018 | 5/23/2018 | 5/25/2018 | 5/28/2018 | 6/4/2018 | 6/9/2018 | |
|------|-----------|-----------|-----------|-----------|-----------|----------|----------|---|
| CF | Rye | 81.8 (27.2) | 38.8 (4.5) | 264.4 (75.4) | 170.8 (28.3) | 166.0 (55.0) | 109.9 (25.2) | |
| | Clover | 38.4 (4.3) | 109.1 (46.3) | 342.0 (93.6) | 517.5 (212.1) | 298.9 (137.7) | 398.8 (213.3) | |
| | Clover-Rye | 38.5 (12.6) | 60.6 (11.2) | 227.5 (22.0) | 166.3 (33.6) | 172.1 (45.6) | 167.7 (39.3) | |
| | Fallow | 17.2 (4.4) | 77.3 (31.1) | 468.3 (153.7) | 275.2 (73.2) | 228.3 (38.9) | 315.3 (51.1) | |
| | | 5/29/2020 | 5/31/2020 | 6/3/2020 | 6/5/2020 | 6/8/2020 | 6/9/2020 | 6/12/2020 |
| KBS | Rye | 117.2 (22.0) | 71.4 (32.0) | 138.7 (27.2) | 112.4 (30.3) | 56.6 (13.2) | 46.2 (17.6) | 57.5 (11.1) |
| | Clover | 53.9 (18.1) | 36.4 (17.0) | 153.6 (83.0) | 150.4 (74.5) | 73.0 (33.1) | 75.4 (13.3) | 57.4 (16.8) |
| | Clover-Rye | 101.1 (49.2) | 44.5 (10.6) | 206.3 (38.1) | 236.0 (69.2) | 125.3 (32.5) | 70.4 (15.9) | 100.1 (22.3) |
| | Fallow | 115.8 (35.5) | 72.9 (30.2) | 265.5 (25.2) | 237.9 (56.8) | 129.6 (26.3) | 98.4 (25.8) | 72.6 (15.2) |

**Table A3: Sensitivity analysis for the *CF* site where we estimated %Ndfa at 40, 50, 60, and 70 for the clover grown alone and in mixture.**

| Treatment | Block | BNF (N kg ha⁻¹) @ 40 %Ndfa | BNF (N kg ha⁻¹) @ 50 %Ndfa | BNF (N kg ha⁻¹) @ 60 %Ndfa | BNF (N kg ha⁻¹) @ 70 %Ndfa |
|---|---|---|---|---|---|
| Clover | 1 | 22.6 | 28.3 | 35.1 | 39.6 |
| | 2 | 44.1 | 55.1 | 68.5 | 77.2 |
| | 3 | 43.9 | 54.9 | 68.1 | 76.8 |
| | 4 | 60.1 | 75.2 | 93.3 | 105.2 |
| | Mean (std. error) | **42.7 (7)** | **53.3 (10)** | **66.3 (12)** | **74.7 (13)** |
| Mixture | 1 | 33.1 | 41.4 | 49.6 | 57.9 |
| | 2 | 32.7 | 40.9 | 49.0 | 57.2 |
| | 3 | 54.0 | 67.5 | 81.0 | 94.5 |
| | 4 | 13.5 | 16.8 | 20.2 | 23.6 |
| | Mean (std. error) | **33.3 (8)** | **41.7 (10)** | **50.0 (12)** | **58.3 (14)** |

755