# Peer review of "Episodic N2O emissions following tillage of a legume-grass cover crop mixture"

_Biogeosciences, 2022_

## Author Response (AR1)

We thank the Associate Editor and two reviewers for your thorough critique and constructive comments on our manuscript. In the sections below we detail how we have incorporated your helpful suggestions into the revised version of the paper. Specifically, we have made changes to clarify the novelty of the paper, better justified our focus on biological N fixation and the clover-rye mixture, added soil mineral N data, estimated cumulative emissions at *CF* if measured for a longer period, and used long-term data from KBS to contextualize $N_2O$ emissions following clover cover crop incorporation at that site. We believe the manuscript is greatly improved thanks to your suggestions and we hope you agree. Thank you once again for your consideration.

**Reviewer 1**

**Section 1:** This manuscript explores short-term N2O emissions after incorporation of cover crops – clover, rye and a clover/rye mix – at two field sites. There are many studies of N2O emissions after incorporation of plant material (cover crops, crop and tree residues), including those that compare legumes vs non-legumes, and so the novelty of the work you present, and the additional knowledge that this provides, are not apparent.

**Response 1:** Thank you for your comment. The novelty of our study is not in comparing legume and non-legume cover crops, but as we discuss in the paper, to our knowledge only two other studies have measured $N_2O$ emissions from fields with legumes as the sole external N input in organically managed grain agroecosystems (**lines 335-340**). Other studies have measured $N_2O$ following legume N inputs + fertilizers/manure, which typically increases overall N inputs compared to our study and does not isolate the effect of legume N sources in organically managed soils. Our study is also one of the first to include measures of labile SOM fractions relevant to internal nutrient cycling processes (i.e., POM) in a study of $N_2O$ emissions. We found one other study (Kong et al. 2009) that measured how soil organic matter fractions and $N_2O$ change over time after conversion to no-till under irrigation in a Mediterranean climate. Based on our review of the literature, our study thus provides new knowledge about episodic $N_2O$ emissions following tillage in agroecosystems with only legume N sources and is also unique for including measurements of SOM fractions.

Kong, A. Fonte, S., van Kessel C. and J. Six. 2009. Transitioning from standard to minimum tillage: Trade-offs between soil organic matter stabilization, $N_2O$ emissions, and N availability in irrigated cropping systems" Soil and Tillage Research 104: 256-262.

**Section 2**: This is confounded by your study being limited in number of spp – legume, non legume and a mix –  and only over a two week period, so that relationships between crop characteristics (eg N content, biomass) or functional traits, can not be rigorously determined, and consequently the discussion provides little insight into trait effects on emissions. I don't consider this limited selection to truly represent 'functional diversity'. You state that little is known about multiple spp, but you are only using one mixture of two spp, and there have been other studies that have measured emissions from these spp, and more rigorously examined effect of spp mixtures.

**Response 2:** Thank you for this feedback. We have double checked that the manuscript does not claim to draw conclusions about the role of functional traits or functional diversity per se, which

was not our intention. In the paper, we simply intend to convey that we increased the functional trait diversity of the main treatment of interest (the two species mixture) by planting a legume and a grass together, which is expected to impact $N_2O$ through effects on plant litter quality and soil N availability. This is better justified now in the introduction with the following language: "In agroecosystems, even small increases in crop functional diversity (e.g., 2-3 species cover crop mixtures with complementary traits) can substantially impact ecosystem functions such as SOM accrual, N cycling processes, and weed suppression (Drinkwater et al., 1998; McDaniel et al., 2014; Tiemann et al., 2015; Blesh, 2017)." **(lines 40-42).** And in the discussion with: "There is growing evidence that small increases in cover crop functional diversity can simultaneously enhance multiple agroecosystem functions, including nutrient retention (Storkey et al., 2015; Blesh, 2017; Kaye et al., 2019). For instance, Storkey et al. (2015) found that low to intermediate levels of species richness (1-4 species) provided an optimal balance of multiple ecosystem services when species exhibited contrasting functional traits related to growth habit and phenology." (**lines 318-322**). We also changed the headings of sections 3.3 (**line 262**) and 4.1 (**line 327**) to specify that we are discussing a legume-grass mixture and not functional diversity more broadly.

Please see the justification for intensively measuring $N_2O$ during the weeks following tillage on **lines 59-68.** We have further supplemented this argument with: "Gomes et al. (2009) found greater $N_2O$ emissions during the first 45 days after terminating cover crops with a roller cutter and herbicide compared to the rest of the year." (**lines 61-62**).

Gomes, J., Bayer, C., Costa, F. D., Piccolo, M. D., Zanatta, J. A., Vieira, F. C. B., and Six, J.: Soil nitrous oxide emissions in long-term cover crops-based rotations under subtropical climate. Soil Tillage Res. 106, 35-44, https://doi.org/10.1016/j.still.2009.10.001, 2009.

**Section 3:** The magnitude of emissions will depend on the chemical composition of the plant material, and this is well established in the literature. The magnitude of emissions from the mixture will depend on the ratio of the component material, and so I find it disappointing that you only applied one ratio of the mix.

**Response 3:** This is an important point, which we now try to better address in the discussion (**lines 388-395**). We note that findings would likely differ for mixtures of different ratios of legume to grass. We conducted our experiment at two sites with contrasting soil fertility levels and did not have the capacity to increase the number of treatments to also test different ratios of legume to grass. However, the strength of our study is that the mixture had similar ratios at both sites, allowing for better comparison of results across sites, and we also achieved a relatively even mixture with strong legume presence, which allowed us to understand the role of fixed N inputs specifically. A growing literature on mixtures argues that mixture evenness is related to agroecosystem multifunctionality, and evenness is thus an important goal of management with mixtures. We have added this point to the discussion. We also note that testing a range of mixture ratios is an important and interesting future research need (**389-390**).

**Section 4:** In the introduction text why do you just focus on emissions from the US? This is a global issue, and by focusing just on the US you are limiting the reach and reader interest of your work.

**Response 4:** Thank you for this suggestion. We have added the following to the beginning of the introduction to provide a more global context: "Globally, $N_2O$ emissions from agricultural soils increased by 11% from 1990 to 2005 and are projected to increase by another 35% between 2005 and 2030 (USEPA, 2012)." (**lines 25-26**)

**Section 5:** Line 53 – 20 years – do you mean 20 days?

**Response 5:** Gelfand et al.'s paper reporting on a long-term study at KBS measured $N_2O$ emissions over 20 years. The point we are making here is that by measuring $N_2O$ over 20 years, the authors found that differences in emissions between years were driven by the episodic emissions immediately following tillage every year, which we use to justify the timing of our sampling to address our research question focused on this particular emissions event following overwintering cover crops.

**Section 6:** Can you please explain why you measured N2 fixation in the legume, rather than just the total biomass N – above + belowground?

**Response 6:** Yes, we added clarification about why $N_2$ fixation is an important aspect of our study to address our question about the role of legume N sources: "Generally, total N inputs are correlated with N losses from agroecosystems (Robertson and Vitousek 2009). However, diversified grain rotations with legume N sources which add biologically fixed $N_2$ to fields, better balance N inputs with harvested exports and have lower potential for N losses compared to synthetic fertilizers (Drinkwater et al., 1998; Blesh and Drinkwater, 2013; Robertson et al., 2014)." (**lines 28-31**)

From an ecosystem perspective, total N inputs to an agroecosystem (regardless of source) are correlated with N losses through leaching or as a gas. For instance, a meta-analysis on $N_2O$ emissions in agroecosystems found that higher total N inputs drive higher $N_2O$ losses by increasing N mineralization (Han et al. 2017) (**lines 366-368**). Legume cover crops add a new N source to soil by fixing atmospheric $N_2$. It is therefore important to partition the legume N into the "new" N, which represents an external input (and is, in principle, more likely to explain loss pathways), compared to N that is assimilated from soil N mineralization and recycled. We also recognize that cover crops (including non-legumes) can increase internal nutrient cycling over time by scavenging and accumulating N and other nutrients in biomass and returning them to soil in relatively labile forms. This dynamic also seems to be a factor in our study and we discuss these processes in the discussion in section 4.2.

**Section 7:** Did you measure changes in soil mineral N after incorporation? I don't see this data, but it will be essential in helping explain the impact on soil processes resulting in emissions, for example net N immobilization (line 317). It is a major omission not to include this data. Likewise, I don't see any measure of CO2 emissions, despite residue addition likely to stimulate microbial activity.

**Response 7:** Thank you for the suggestion to include soil mineral N changes. We did measure this and have added the methods in section 2.6 (**lines 184-189**) and added Table 2 with soil inorganic N data at two different time points at each site (on the day after tillage, and 12-13 days

later) and added a summary this to the results **(lines 219-221).** We have also added this component throughout the discussion **(lines 349-351; 412-413; 430)** We did not measure $CO_2$ emissions in this study but agree that microbial activity increased with the addition of fresh cover crop residue (e.g., as shown by the release of inorganic N and flux of $N_2O$ measured in our experiment).

**Section 8:** I may have missed this, but I don't see data of the chemical characteristics of the clover, rye or weeds?

**Response 8:** In Figure 1 and section 3.2 of the results, we report on litter N and C:N ratio for all cover crop treatments. We did not include other measures of litter chemistry (e.g., lignin) in this study.

**Section 9:** It would be helpful to have the daily fluxes of N2O also presented as fluxes per biomass or % C applied basis. I think you give this for cumulative N2O, but not for the daily fluxes.

**Response 9:** Thank you for this suggestion. We have added table in the appendix (Table A2) with daily $N_2O$/aboveground cover crop biomass and biomass N. We have referenced this new table in the text on **lines 308-309**.

**Reviewer 2:**

**Section 1:** The manuscript is well written and assessing interesting question regarding the effect of cover crop (and mixtures) incorporation on soil nitrous oxide emissions. At KBS site measurement length seems to be appropriate for the question asked, at the CF site, however, the post-incorporation peak emissions have not finished before the last measurement (e.g. Fig 2). Thus, cumulative emissions calculated for the CF cycle likely underestimated. This problem can be addressed, at least partially by additional analysis of the existing data.

**Response 1.** Thank you for your positive feedback on the manuscript, and for this helpful comment regarding the data at *CF*. We agree this is the case for the clover treatment at *CF*. We now better acknowledge in the discussion that the estimate for cumulative emissions at *CF* is likely an underestimate. We add an analysis in **lines 380-386** to provide a possible range of cumulative emissions for this treatment had we measured for a longer period: "Differences between site were likely even higher than our data suggests. We likely underestimated cumulative $N_2O$ emissions during the first peak following tillage at *CF* because emissions had not yet returned to baseline, especially for the clover treatment. By extending our empirical measurements using regression models, we estimated the trajectory of $N_2O$ emissions to approximately 19-26 days after tillage depending on the cover crop treatment and replicate. We estimate that cumulative $N_2O$ emissions at *CF* could have reached $822.8 \pm 253.2$ g $N_2O$ N ha$^{-1}$ in clover, $461.6 \pm 59.2$ g $N_2O$ N ha$^{-1}$ in mixture, $340.4 \pm 63.4$ g $N_2O$ N ha$^{-1}$ in rye, and $355.0 \pm 77.4$ g $N_2O$ N ha$^{-1}$ in fallow. These higher estimates further increase differences in cumulative $N_2O$ emissions between sites."

**Section 2:** I think that authors should include analysis of post-incorporation emissions from the KBS LTER site since, I guess CF site doesn't have long-term soil $N_2O$ emissions data. Within

existing data authors can find times of cover-crop incorporation across the KBS dataset. By finding measurements of post-incorporation emissions and compare them to total annual/seasonal emissions, authors can prove that post-incorporation emissions indeed contribute significant amount of $N_2O$ emissions. This will improve the manuscript and make it more suitable for publication. I agree with the first reviewer comments and don't want to repeat them, however, I think that incorporation of additional analysis will make this manuscript suitable for publication, despite limited novelty pointed by the reviewer 1.

**Response 2:** Thank you for this great suggestion! Based on historical $N_2O$ data at the KBS site, we analyzed $N_2O$ emissions when they were measured within four weeks following incorporation of the red clover cover crop in the organically managed treatment at KBS. We now report this to add additional context to our short- measurements in the discussion: "Additionally, we used long-term measurements of $N_2O$ emissions from the biologically-based cropping system at KBS as further context for interpreting our single-season results. Between 2014 and 2020, following the red clover cover crop, there were three years in which $N_2O$ fluxes were measured roughly two weeks apart within a month after tillage. These two-week periods of $N_2O$ emissions after tilling red clover represented 19.9 ± 2.1 % of the annual emissions from this cropping system (Robertson 2020). These $N_2O$ measurements from past years at the KBS site were not collected until at least 8 days after tillage, and likely missed the initial flux immediately following soil disturbance, which may explain why we found a slightly higher proportion of annual emissions (26.3%) following clover tillage at *KBS*. These historical data suggest that we indeed captured the peak $N_2O$ flux following soil disturbance by tillage in our one-year experiment." (**lines 461-468**).

Robertson, G.: Trace Gas Fluxes on the Main Cropping System Experiment at the Kellogg Biological Station, Hickory Corners, MI (1991 to 2019) ver 46, Environmental Data Initiative, https://doi.org/10.6073/pasta/b1feb30692eb31b7f8a27615d18e0fa8 (Accessed 2022-02-11), 2020.

**Section 3:** Two technical comments: 1. please use appropriate decimal numbers in current version you use non, one, and two decimal numbers sometimes in the same paragraph (L237, section 3.2). 2. Figure 2, please do not use smoothing line or connection line - you have not measured continuously.

**Response 3:** Thank you for picking up on the inconsistency. We have checked for decimal places to be consistent and made edits throughout the manuscript. We appreciate this comment about the smoothing line but would also argue there are differing opinions on the acceptability of this approach in the literature on $N_2O$ emissions. In this case, the smoothing lines greatly improve the visualization of the patterns between treatments and across sites and we prefer to keep them in. We explain in the methods exactly how we calculated/estimated this curve (see Eq. 2 on **line 197**). We have added a section on the limitations of this approach after the equation in the methods: "In the absence of continuous sampling, this approach allowed us to approximate a total flux over the sampling window and better visualize treatment patterns within and across sites." (**lines 199-200**) We will also add a note to the Figure 2 caption: "The lines connecting the sampling points are intended to aid in visualizing treatment patterns for cumulative $N_2O$ and do not indicate continuous data collection (Eq. 2)." (**lines 729-730**)